

# Mining integrated semantic networks for drug repositioning opportunities

Joseph Mullen[1], Simon J. Cockell[2], Hannah Tipney[3],
Peter M. Woollard[3] and Anil Wipat[1]

[1] Interdisciplinary Computing and Complex BioSystems Research Group, School of Computing Science, University of Newcastle-upon-Tyne, Newcastle upon Tyne, United Kingdom
[2] Bioinformatics Support Unit, University of Newcastle-upon-Tyne, United Kingdom
[3] Computational Biology, Target Sciences, GSK R&D, GlaxoSmithKline, Stevenage, Hertfordshire, United Kingdom

## ABSTRACT

Current research and development approaches to drug discovery have become less fruitful and more costly. One alternative paradigm is that of drug repositioning. Many marketed examples of repositioned drugs have been identified through serendipitous or rational observations, highlighting the need for more systematic methodologies to tackle the problem. Systems level approaches have the potential to enable the development of novel methods to understand the action of therapeutic compounds, but requires an integrative approach to biological data. Integrated networks can facilitate systems level analyses by combining multiple sources of evidence to provide a rich description of drugs, their targets and their interactions. Classically, such networks can be mined manually where a skilled person is able to identify portions of the graph (semantic subgraphs) that are indicative of relationships between drugs and highlight possible repositioning opportunities. However, this approach is not scalable. Automated approaches are required to systematically mine integrated networks for these subgraphs and bring them to the attention of the user. We introduce a formal framework for the definition of integrated networks and their associated semantic subgraphs for drug interaction analysis and describe DReSMin, an algorithm for mining semantically-rich networks for occurrences of a given semantic subgraph. This algorithm allows instances of complex semantic subgraphs that contain data about putative drug repositioning opportunities to be identified in a computationally tractable fashion, scaling close to linearly with network data. We demonstrate the utility of our approach by mining an integrated drug interaction network built from 11 sources. This work identified and ranked 9,643,061 putative drug-target interactions, showing a strong correlation between highly scored associations and those supported by literature. We discuss the 20 top ranked associations in more detail, of which 14 are novel and 6 are supported by the literature. We also show that our approach better prioritizes known drug-target interactions, than other state-of-the art approaches for predicting such interactions.

Corresponding author
Anil Wipat, anil.wipat@ncl.ac.uk

## INTRODUCTION

Drug repositioning is the process of finding new uses for existing drugs. This process is a rapidly-evolving issue in the area of drug development, having the potential to reduce both drug development costs and the time taken for a drug to reach the market. Many repositioned drugs currently on the market have been discovered through either serendipitous or rational observations. However, these manual approaches are not efficient given the potentially huge search space of drug-target (D-T) interactions. Systematic approaches to searching for repositioning opportunities are required to provide an efficient and scalable alternative to manual investigations.

A large number of studies have detailed computational approaches to aid in the systematic identification of drug repositioning opportunities, including methodologies based on: chemical structure (*Keiser et al., 2009*), protein structure and molecular docking (*Moriaud et al., 2011*), phenotype similarity (such as side-effect similarity (*Yang & Agarwal, 2011*) and gene expression similarity (*Lamb et al., 2006*)) or genetic variation (*Sanseau et al., 2012*). Approaches aim to infer links in the drug-target-phenotype-disease schema (*Hurle et al., 2013*). For example, side-effect methods link a known drug-phenotype to a new disease (drug-phenotype-disease). Genetics-based methods, however, can link targets with a phenotype that is associated with the disease (drug-target-phenotype-disease) (*Hurle et al., 2013*). One may also focus on the prediction of drug-target associations, with the hope that hypothesised links generated from domain knowledge will allow us to complete a drug-target-disease pathway and infer a novel use for an existing drug. As well as highlighting potential drug repositioning opportunities, D-T interaction identification also allows potential adverse side effects to be analysed (*Fakhraei et al., 2014*; *Dudley et al., 2011*).

In vitro approaches to identifying D-T interactions are no different to other aspects of drug development and remain costly and time consuming (*Ding et al., 2014*). Using systematic *in silico* prediction methods allows for the D-T interaction search space to be reduced, highlighting areas for focus (*Fakhraei et al., 2014*). Molecular docking methodologies are heavily applied to the task, but require a large amounts of computational resources and are time consuming (*Ding et al., 2014*). Other approaches involve machine learning-based methods which may utilise a feature vector approach or, more commonly, similarity-based approaches which exploit the similarity between drugs and proteins (*Ding et al., 2014*). Such approaches allow for the production of prediction models and can be ligand-based or structure-based. For example, ligand information may be used to create models that learn which sub-structural features of a ligand correlate with activity against a particular target (*Alvarsson et al., 2014*). Other similarity-based approaches make use of a network, or more specifically a bipartite graph, data representation (*Ding et al., 2014*; *Fakhraei et al., 2014*; *Palma et al., 2014*; *Yamanishi et al., 2008*; *Yamanishi et al., 2010*). Within a bipartite graph, vertices are divided into two disjoint sets, proteins and drugs. Data from multiple publicly accessible datasets is integrated during the building of these networks (*Lee et al., 2009*), yet in most approaches to D-T interaction prediction data is limited to the inclusion of the two data 'types', protein and drugs.

More recent approaches to drug repositioning focus on the creation of integrated networks which combine data from multiple analyses, to give a systems level view of cellular and molecular processes (*Barratt & Frail, 2012*; *Chen et al., 2012*; *Cockell et al., 2010*; *He et al., 2011*; *Iskar et al., 2012*). This approach provides a complementary path to reductionist science in understanding complex phenomena. Semantically-rich integrated networks, which utilise a graph-based representation, are a convenient method of representing the types of integrated data necessary for finding drug repositioning opportunities (*Betzler et al., 2011*). In graph-based data, entities, such as proteins or drugs, are represented as vertices. Interactions between these entities, such as protein-protein interactions or a drugs binding to a protein are captured in edges. In *semantic graphs* each vertex and edge in the graph is assigned a type from a predefined set. Vertices and edges are also are annotated with attributes. Graph representations of complex systems are widely used in computer science, social and technological network analysis science due to their ability to represent structured and semi-structured data (*Riaz & Ali, 2011*). Within bioinformatics graph-based representations are also widely adopted, particularly as a means of representing data produced during an exercise in data integration and in protein-protein interactions networks.

In the context of these integrated networks, subgraphs are connected components of the parent network (*Gallagher, 2006*). These subgraphs formally capture local relationships between the elements represented in the graph. Often, the relationships in a given subgraph are indicative of a particular biological phenomenon. In the case of drug repositioning networks, the types of relationships include amongst others: interactions between drugs and their targets, interactions between targets, and the diseases associated with particular targets. Therefore, within the integrative graph are subgraphs that describe repositioning opportunities as a result of their semantic and topological properties. Once appropriate subgraphs have been observed and defined they can be used as templates to find instances of these subgraphs, and related subgraphs, within a given graph to highlight similar drug repositioning opportunities.

For example, chlorpromazine is an anti-psychotic drug that is also approved as an antihistamine (*Mitchell, 1993*). The interactions of chlorpromazine can be captured in an integrated network (Fig. 1). Data from DrugBank version 2.5 (DBv2.5) (*Wishart, 2006*) provides three interactions between chlorpromazine and single protein targets; none of these interactions explain the antihistaminic affects of the drug. Structurally, chlorpromazine is very similar to the antiemetic trimeprazine. DBv2.5 captures an interaction between trimeprazine and the Histamine H1 receptor, a known target for antihistamines. Through guilt-by-association, we can therefore predict the Histamine H1 receptor as a target for chlorpromazine, an interaction captured in the latest editions of the DrugBank database. The topological and semantic properties of the subgraph depicted in Fig. 1B describe a repositioning relationship that could be generically applicable to any two drugs and their target. Fig. 1B describes a situation whereby a compound, structurally similar to a compound with a known target, may also bind to the same target (the inference is represented as the dashed line). This real example can therefore be used to derive a template semantic subgraph that can be used for searching
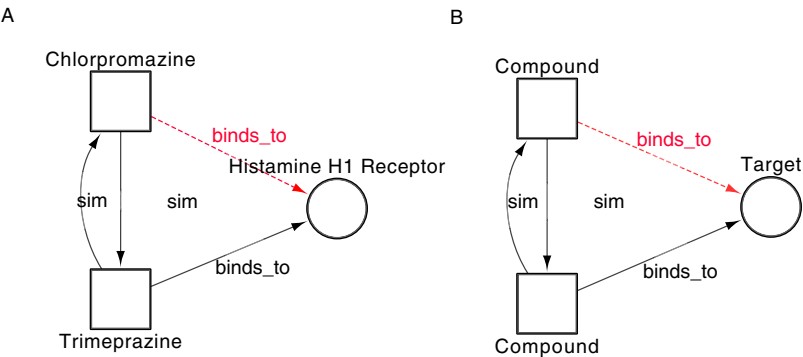

**Figure 1 An example of a simple semantic subgraph (B) is derived from the repositioning of Chlorpromazine (A).** Chlorpromazine is marketed as a non-sedating tranquilliser, but is also known to be effective as an antihistamine (*Rukhadze et al., 2001*) and so in A a relation is inferred between Chlorpromazine and the Histamine H1 receptor (dashed line).

for similar, but novel, drug-target associations relationships involving different drugs and targets. This template semantic subgraph therefore describes a pattern indicative of a drugs interaction with a target, highlighting potential new indications for the drug. Although Fig. 1 shows a simple triad, semantic subgraphs capturing data relevant to repositioning opportunities are likely to be more complex. In the context of drug repositioning, manual identification of potential repositioning opportunities from large target networks is possible, though not efficient for the systematic analysis of such large networks. The definition of semantic subgraphs for known repositioning opportunities, in combination with an algorithm for the mining of integrated complex networks for these subgraphs, allows us to highlight potential repositioning in a more systematic and exhaustive fashion.

In this paper we introduce a formal framework for the definition of a semantic subgraph for integrated networks. We also present DReSMin (**D**rug **Re**positioning **S**emantic **Min**ing), an algorithm for searching integrated networks for occurrences of a given semantic subgraph using semantic distance thresholds. DReSMin optimises the search time for larger subgraphs by including a semantic graph pruning step and applying a method for splitting large subgraphs prior to searching. We demonstrate the utility of our approach by searching an integrated drug dataset for semantic subgraphs that are indicative of drug repositioning opportunities, particularly focusing on inferring D-T interactions. As part of this work we updated an existing integrated dataset used for *in silico* drug discovery (*Cockell et al., 2010*). Finally we demonstrate that our approach can be successfully used to predict putative D-T interactions that were not explicitly represented in the integrated network.

## Graphs

### Definition of our graph model

A graph $G$ is defined as a ordered pair $(V,E)$, where $V$ is a set of vertices (or nodes), and $E \in V \times V$ is a set of edges (or relations). Each $e \in E$ is a pair $(v_i, v_j)$ where $v_i, v_j \in V$.

Edges represent relations between vertices. Edges may be directed or undirected. Both vertices and edges may be labelled, typed and attributed.

DReSMin, requires a *directed* (edges have a direction associated with them) graph where vertices and edges are labelled with types $T_v$ and $T_e$ respectively, where $T_v$ and $T_e$ are drawn from a finite hierarchy of types $H$, and are annotated with attributes. The algorithm allows for *multigraphs* (vertices $v_i$ and $v_j$ are permitted to have multiple edges between them) and for vertices to contain *self-loops* ($v_i$ may have an edge directed toward itself). For the remainder of this paper $|V(G)|$ will be used to represent the number of vertices contained in graph $G$.

### Classical subgraph definition

Subgraph isomorphism is a task in which two graphs, $G$ & $Q$ are given as input and one must determine whether $G$ contains a subgraph that is isomorphic to $Q$: is there a subgraph $G'(V',E')$: $V' \subseteq V, E' \subseteq E$? During the search of a query graph, a mapping ($M$) is expressed as the set of ordered pairs $(v,m)$ (with $v \in G$ and $m \in Q$) and so $M = \{(v,m) \in V_G \times V_Q | v$ is mapped onto $m\}$; that is $M: G' \mapsto Q$.

### Semantic subgraph definition

A semantic subgraph is defined as $Q = (V, E, T_v, f_v, T_e, f_e)$, where $V$ is a set of vertices, $E$ is a set of edges, $T_v$ is a set of node types and $T_e$ is a set of edge types. $f_v : V \twoheadrightarrow T_v$ and $f_e : E \twoheadrightarrow T_e$ are surjective functions; each node is assigned a node type and each edge an edge type from $T_v$ or $T_e$ respectively. A semantic subgraph may be designed in such a manner that mappings, or occurrences, in $G$ aid in the inference of a relation between vertices of a particular $t_e$, where a relation does not exist. For example, one may use the semantic subgraph depicted in Fig. 1B to infer an interaction between a compound and a target.

## Graph matching

Several approaches have been described for combining semantic information with network motif topology including the *list coloured motif problem* (*Betzler et al., 2011*; *Lacroix et al., 2006*). In this case a motif ($M$) is defined as a multiset of colours, or types. An occurrence of $M$ is a subset of vertices that forms a connected subgraph whose multiset of colours, or types, matches that of $M$ exactly (*Lacroix et al., 2006*). Although this approach demonstrates how network motifs may be extended to incorporate semantic information, it does not allow for topological exacts to be identified. The ability to identify sub-components of a target network that match a defined topology is a necessity in situations where the topology of a subgraph is believed to aid in describing the functionality of the sub-component. The task of identifying mappings of a predefined subgraph with similar topology from a larger graph is known as the *graph matching problem* (*Gallagher, 2006*).

There are different variations of the graph-matching problem. For example, *exact matching* occurs when the mapping between the vertices of the two graphs is *edge-preserving*; a mapping contains all edges defined by the query. One of the most stringent forms of exact matching is *subgraph isomorphism* (*Conte et al., 2004*) which aims

to find all occurrences of a query graph and is an NP-complete problem (*Washio & Motoda, 2003*). Currently, algorithms addressing this problem are exponential in performance relative to the size of the input graphs (*Gallagher, 2006*). Many algorithms which have been developed to address the subgraph isomorphism problem are based on the exhaustive algorithm developed by *Ullmann (1976)*. Applying an exhaustive method to the identification of drug repositioning opportunities is important to ensure all possible novel applications for a drug are investigated. Using a backtracking approach, Ulmann's algorithm finds solutions by incrementing partial solutions or abandoning them when determining they cannot be completed (*Ullmann, 1976*). An extension of the Ullman approach, incorporating the semantics of a graph, has been implemented using inexact (*Djoko et al., 1997*), as well as exact approaches (*Cordella et al., 2004*; *Giugno & Shasha, 2002*). However, as yet, none of these approaches have been applied to the automated identification of drug repositioning opportunities.

Whilst searching for semantic subgraphs it is important to consider the similarity between the query subgraph and the target, both in terms of graph topology and the meaning of the annotations on vertices and edges. A measurement of semantic similarity between elements of a mapping and the equivalent element in a query must be introduced to the search and the degree of similarity can be expressed as a semantic distance. Numerous measures have been developed to score the semantic similarity between two ontological concepts (*Ge & Qiu, 2008*; *Noy, 2004*). Previous work in the area of intelligence link analysis has used ontology-based semantic similarity scoring methods for pattern matching (*Seid & Mehrotra, 2007*). In Seid and Mehrotra's algorithm, an inexact topological search is carried out with matches semantically scored based on their Least Common Ancestor (LCA) within an ontology. Topological and semantic scores are then combined and $k$ ranked matches returned.

Whilst approaches described are adequate for their particular setting, here we present a new exhaustive graph matching approach to aid in the identification of potential drug repositioning opportunities from a target network. We therefore describe an algorithm for this task which is an improvement on those introduced for the purpose of drug repositioning.

## MATERIALS AND METHODS

### Algorithm

We have developed DReSMin, an algorithm for the detection of semantic subgraphs. This algorithm returns all mappings of a semantic subgraph that match at a level equal to, or above a given threshold, *ST*. In this case our application for the algorithm is the identification of a semantic subgraph ($Q$) which may be indicative of drug repositioning opportunities within a target graph ($G$). Examples of semantic subgraphs may be initially drawn from a set of templates, that is the graph representation of known repositioned drugs, such as chlorpromazine, shown in Fig. 1A. The algorithm is made up of four main components which are described in Fig. 2. These components comprise: (i) Semantic graph pruning (ii) Topological search (iii) Semantic subgraph distance exclusion (iv) Semantic subgraph splitting.

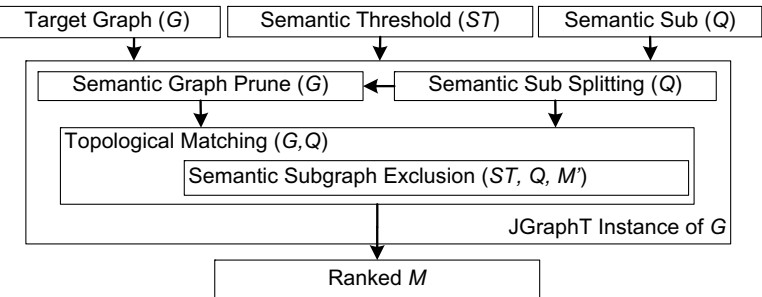

**Figure 2 Overview of the DReSMin algorithm developed for the detection of semantic subgraphs indicative of repositioning opportunities.**

## Semantic graph pruning

We are concerned only with identifying semantic subgraphs that match, semantically, at a level equal to, or above our threshold. (Note: In this work, the semantic distance between two graph entities is calculated using the semantic distance calculator described in the 'Semantic subgraph distance exclusion' section.) In this graph pruning component of the algorithm, any vertices (and their associated edges) in $G$ which are above a certain semantic distance from those in $Q$ are removed from $G$. This step allows any vertices that are semantically distant from our query to be removed prior to a search, cutting down the search space. Taking $G$, $Q$ and a semantic threshold ($ST$) each $t_v \in T_v(Q)$ are sent to the semantic subgraph distance calculator (termed SDC and described later in the manuscript), and scored against every $t_v \in T_v(G)$. If $SDC(t_v(Q), t_v(G)) < ST$ then all $v \in V(G)$ of type $t_v$ are removed from $G$ as well as any $e \in E$ where $v = v_i$ or $v = v_j$. Finally after all semantically insignificant elements are removed from $G$, all isolated $v \in V(G)$ that may have resulted from the edge pruning step are also removed.

## Topological matching

Many algorithms addressing the problem of subgraph isomorphism build on Ullman's work. These applications include: GraphQL (*He & Singh, 2008*), GADDI (*Zhang, Li & Yang, 2009*) and, one of the most efficient, the VF algorithm (*Cordella et al., 1999*). Performance is increased in these algorithms by exploiting different join orders, pruning rules and auxiliary information to prune out negative candidate subgraphs as early as possible. We carry out topological matching using a variation of the VF algorithm (*Cordella et al., 1999*). The VF algorithm is exhaustive and suitable for working with 'large' graphs (up to $3 \times 10^4$ vertices) and employs a depth-first strategy implemented in a recursive fashion (*Cordella et al., 1999*). During a search using the VF algorithm, the search space is minimised via the introduction of topological pruning rules (*Cordella et al., 1999*). Integrated networks typically surpass the aforementioned 'large' graphs in size, particularly true within the biological and pharmaceutical settings. As data volumes continue to grow (e.g. omics technologies continue to mature) it is important to develop exhaustive algorithms capable of scaling with the data.

Our initial implementation of the VF algorithm showed poor scalability and so, as an enhancement to the VF algorithm, we introduce three steps to improve the efficiency of searching for topological subgraphs. These three steps include: a set of rules used to determine the appropriate vertices at which an instance of the search is started (initial candidate set), as described in (1) below; a topological pruning rule, based on a closed world assumption, as described in (2) below; and a semantic thresholding step (described in the next section of the manuscript). As the focus of this work is the inference of associations involving compounds, it is vital that all mappings resulting from a search contain a node of this particular semantic type.

1. When considering an initial candidate set of nodes from the target graph $G$ at which to initiate the search, it is desirable to try to ensure that the set is made up of nodes of a type, $X$, such as `Compound` to ensure the relevance of the portion of graph being searched. Therefore, an initial candidate set for the search is chosen based on: all $v \in V(Q)$ whose $t_v \in T_v(Q) = X$ are considered with $v > deg_Q(v)$ (where $deg$ represents the degree of a node) selected as $v$. $m$ is made up of all $v \in V(G)$ whose $deg_G(v) \geq deg_Q(v)$ and $t_v \in T_v(G) = X$.

2. When mining with a given semantic subgraph that describes a potential repositioning situation we must assume that the lack of a relationship between nodes indicates the absence of a relationship between the two nodes (a closed world assumption). As a result, when searching for a given semantic subgraph, $Q$, we only consider a match if there exists no additional edges between the vertices in a mapping $M$ from the target graph $G$, and their equivalent vertices in $Q$. Therefore, a mapping $M$ is expressed as a set of ordered pairs and the closed world assumption requires $(M = match) \lor (deg(v) \in (G) \equiv deg(m) \in V(Q))$.

### Semantic subgraph distance exclusion

Semantic thresholding is used to exclude matches found in $G$ that are below a given semantic distance from $Q$. This process is achieved through a semantic subgraph distance calculator (SDC). An SDC comprises of two distance matrices, one for $t_v \in T_v(G)$ and one for $t_e \in T_e(G)$. We have $n = 19(t_v)$ and $n = 42(t_e) => $ each matrix is represented as matrix $P' = (p_{ij})$, the $n \times n$ matrix defined by;

$$p_{ij} = \begin{cases} 1 & \text{if } p_i \text{ is semantically identical to } p_j; \\ 0 & \text{if } p_i \text{ is semantically unrelated to } p_j; \\ -1 & \text{if } p_i \text{ is semantically opposite to } p_j. \end{cases} \quad (1)$$

During the matching process each element of $M = (V_m, E_m)$ is scored against its equivalent in $Q = (V_s, E_s)$. The resulting *semantic score* (SS) of $M$ is;

$$\sum \frac{SDC(m_1, q_1), SDC(m_2, q_2)....SDC(m_n, q_n)}{n} \quad (2)$$

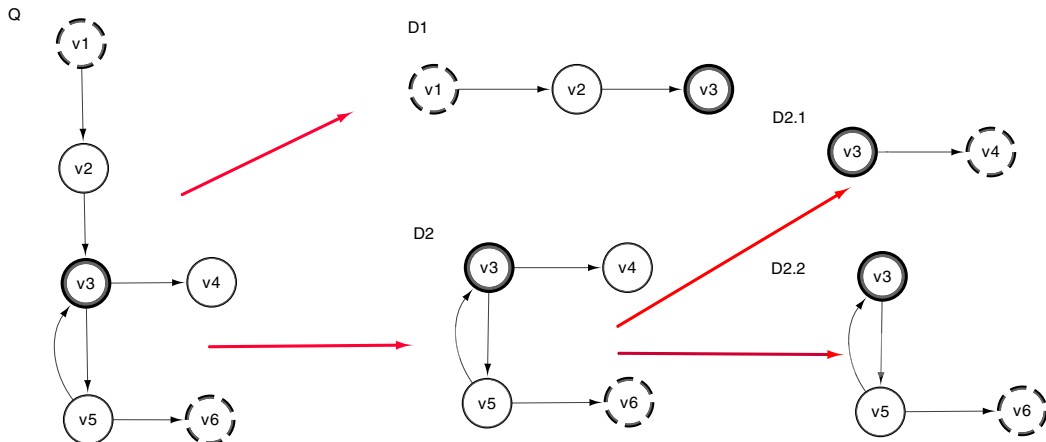

**Figure 3 Subgraph split procedure takes an initial semantic subgraph ($Q$) and produces two smaller semantic subgraphs ($D_1$ and $D_2$) using all vertices ($v$) from ($Q$).** The overlapping node ($ON$) is identified in $Q$($v_3$) and used as the overlapping node in both $D_1$ and $D_2$. The two most distant vertices in $Q$ are then identified ($v_1$ and $v_6$) and vertices in the path between these and $ON$ added to the corresponding graphs ($D_1$ and $D_2$). We also see that $|V(D_2)| > 3$ and so a second call is made to graph split giving us $D_2 1$ and $D_2 2$.

A semantic threshold ($ST$) is defined by the user prior to a search; a value ranging from 0 to 1. During the search, vertices and edges pass or fail the semantic threshold. Thus we identify topological exacts and semantic closeness.

### Semantic subgraph splitting

This component takes a semantic subgraph, $Q$, and returns a set of semantic subgraphs, $D$, whose $|V| < 4$. In Fig. 2, we see how this step interacts with the other components of DReSMin. $\forall d \in D$ produced during this step of DReSMin the target network, $G$, is pruned using the semantic graph prune component and $d$, before $d$ is searched for in $G$. The graph splitting component allows smaller subgraphs to be searched and mappings joined based on sharing a common overlapping node ($ON$). In order for this approach to be successful a semantic subgraph is first converted to an undirected graph. The most connected node, $v_{max}(Q)$, is then identified and used as $ON$. Of all the remaining $v \in V(Q)$, the two most distant vertices ($v_1$, $v_2$) from $Q$ are selected. Two new graphs ($D_1$ & $D_2$) are then created and populated with nodes as such: $V(D_1) \cup v \in \delta(v_1, ON)$, $V(D_2) \cup v \in \delta(v_2, ON)$, that is every node in the shortest path from $v_1$ to $ON$ is included in $D_1$ and every node in the shortest path from $v_2$ to $ON$ is included in $D_2$. Remaining vertices are then allocated depending on which graph they share a connecting edge with Fig. 3. Edges are then allocated as such: $\forall e \in E(Q)$ if either $V(D_1)$ or $V(D_2)$ contains both ($v_i$, $v_j$) of $e$; $e$ is allocated to that graph. Any edges whose nodes are are not found in the same graph are not allocated to the split subgraphs. As a result of this process during a search we have $D_1$ and $D_2$ as well as our original semantic subgraph, $Q$. A search is then started with $D_1$ or $D_2$, depending on which has the smallest $|V|$. The search is started using $ON$, maintaining the edge set it possessed in $Q$, reducing the initial candidate set. All starting vertices that lead to an embedding being identified are then passed to the

second search; reducing the initial candidate set once more. All matches from the two searches who share a common *ON* are then mapped and a final check for any $e \in E(Q)$ that were not allocated to either $D_1$ or $D_2$ is made. This splitting may be called iteratively if either $D_1$ or $D_2$ still possess a $|V| > 3$ after the first round of splitting, as demonstrated in Fig. 3. Subsequent searching will result in the same set of mappings that would be identified by a non-split search (for algorithm pseudo-code and discussion please see Article S1).

### Ranking inferred interactions

Scoring of a semantic subgraph, $Q$, is achieved by determining the number of known D-T interactions in the predicted total set of D-T interactions inferred by $Q$. We refer to the complete set of inferred interactions as $Q(I)$. A score $R_q$ is calculated based on the ability of $Q$ to identify D-T interactions captured in DBv3, but not present in our *Dat* integrated data set (see next section). The set of interactions that are captured in DBv3, but not captured in *Dat* is known as *DBv3Rel* (Eq. 5).

$$R_q(Q) = \frac{|Q(I) \cap DBv3Rel|}{|Q(I)|} \tag{3}$$

Once $R_q$ is calculated for each semantic subgraph we then score individual D-T interactions, $i$, based on the cumulative score of all semantic subgraphs that predicted $i$.

$$R_i(i) = \sum_{i \in Q'(I)} R_q(Q') \tag{4}$$

DReSMin is an exhaustive algorithm, as such, scoring inferred interactions allows for ranking, with those ranked higher inferred with greater confidence than others.

### Characterisation and application

An integrated dataset for *in silico* drug discovery has been described previously by Cockell and co-workers (*Cockell et al., 2010*). This dataset satisfies the requirements described for our algorithm (see 'Definition of our graph model' section) and so was used to test the algorithm performance and mined for D-T interactions using a Java based implementation of DReSMin.

The dataset was developed in Ondex (*Köhler et al., 2006*) and includes compounds and targets from DrugBank[1] (*Wishart, 2006*), Proteins from UniProt[2] (*UniProt Consortium, 2013*) as well as information from eleven other databases and analysis methods (*Cockell et al., 2010*). An updated version of this dataset was used as a test bed for this work, however the approach we describe is valid for most integrated networks that adopt a semantically rigorous approach to edge and vertex type definition.

Utilising a graph-based data representation and providing a framework for visualisation, both vertices and edges within an Ondex graph are annotated with semantically enriched metadata. Each vertex (or concept) is assigned a $c \in C$, where $C$ is a finite set of `conceptClasses`, while each edge or relation is assigned a $r \in R$ where $R$ is a finite set of `relationTypes` (*Köhler et al., 2006*). As part of this work we developed

[1]http://www.drugbank.ca.

[2]http://www.uniprot.org.
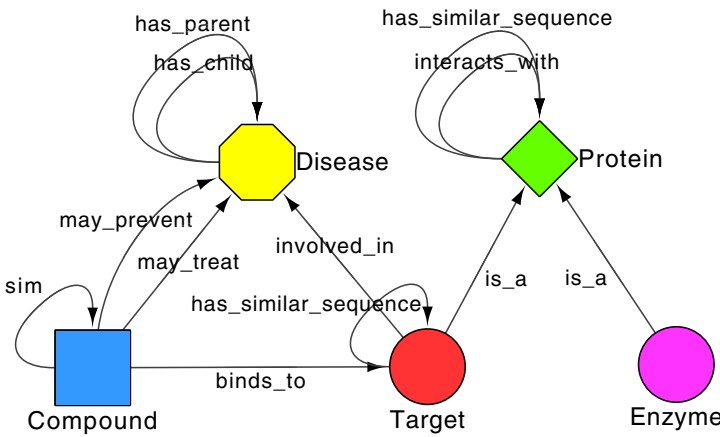

**Figure 4 A subsection of the Ondex *in silico* drug discovery dataset metagraph.** Shows how different `conceptClasses` (E.g. Compound & Target) interact via `relationTypes` (`binds_to`).

[3]VA National Drug File Reference Terminology. <http://www.nlm.nih.gov/research/umls/sourcereleasedocs/current/NDFRT>
Accessed September 2013.

[4]Gene-disease association data were retrieved from the DisGeNET Database, GRIB/IMIM/UPF Integrative Biomedical Informatics Group, Barcelona. <http://www.disgenet.org/>.
Accessed September 2013.

[5]https://www.ebi.ac.uk/chembl/.

plug-ins (parsers and mappers) for the Ondex platform to extend the original dataset. These plug-ins allowed us to add `disease conceptClass`, taken from the National Drug File Reference Terminology (NDF-RT)[3]. Four `relationTypes` showing interactions between `Disease-Disease` (`has_parent` and `has_child`) and `Compound-Disease` (`may_treat` and `may_prevent`) originally defined in NDF-RT were also integrated. A final `relationType` between `Target-Disease` (`involved_in`) was integrated from DisGeNET[4] (*Bauer-Mehren et al., 2010*). The updated dataset, which we refer to as *Dat* from here on in, has an additional 4,463 vertices (155,316) made up of 19 `conceptClasses` (see Table S1) in comparison to the original, together with an additional 28,736 edges (816,096), representing 42 `relationTypes` (see Table S2). The metagraph of the dataset described is shown in Fig. S1, with a subsection shown in Fig. 4.

This graph shows a high degree of connectivity with a $d_S(G)$ (average node degree) of 10.42, whereby degrees of vertices range from $\delta(G)$ (minimum degree) of 1 and $\Delta(G)$ (highest degree) of 15,004. Average connectivity differs between `conceptClasses`, with `Proteins` displaying the highest $d_S(G)$ of any `conceptClass` at ~45. Other notable connectivity averages include `Target` ~13, `Compound` ~7 and `Disease` ~4. All searches presented here were carried out using a semantic threshold ($ST$) of 0.8 (see Article S2). We only include vertices of type `Compound` in our initial candidate set.

### Drug-Target interaction prediction evaluation

We compared our ranked set of predicted D-T interactions to those produced by another state-of-the-art method for drug target interaction prediction–a ligand-based method. One implementation of such an approach is provided by ChEMBL[5]. ChEMBL provide two models for target prediction, using bioactivity data with a cut-off of 1 $\mu$M and 10 $\mu$M respectively. These models allow for n predicted interactions to be made for a given drug. Inferred interactions are also scored and can be ranked, meaning a direct comparison to our approach can be achieved. Predictions using the ChEMBL models can be found in compound report cards, accessed via their website.
[6] www.ebi.ac.uk/unichem/ Accessed 22nd June 2015.

[7] https://github.com/chembl/chembl_webresource_client.

[8] x = 100 or, if DReSMin inferred <100 targets for this drug, x = number of DReSMin inferred targets.

[9] http://www.uniprot.org Accessed July 30th 2015.

[10] http://www.uniprot.org/docs/7tmrlist. txt Accessed Nov 11th 2015.

[11] http://www.uniprot.org/docs/pkinfam. txt Accessed Nov 11th 2015.

[12] http://www.uniprot.org/docs/peptidas. txt Accessed Nov 11th 2015.

Mappings between DrugBank and ChEMBL compounds were retrieved from UniChem (*Chambers et al., 2013*) via whole source mapping[6]. This mapping provides a set of 3,765 drugs that are contained in both datasets, of which 57 of the ChEMBL ids mapped to >1 DrugBank ID (one to four, five to three, and 51 to two). DReSMin inferred D-T associations for 2,223 of drugs common to both databases. In the comparison presented below we only consider D-T interaction inferences involving this set of 2,223 drugs. The set of inferences from DReSMin contained a total of 2,456 protein targets (of which 1,133 are from *Homo sapiens* and 1,323 from other organisms). The set of ChEMBL inferences involve 870 human protein targets, of which 362 are also captured in DReSMin inferred D-T associations.

For each of the 2,223 drugs, we identified associations with single proteins. The top 100 of these associations were identified using the ChEMBL Web resource client[7]. Any interactions which were already captured in *Dat*, involved targets from organisms other than humans or were not captured in the overlapping 362 protein targets, were excluded from the analysis. This process was repeated for both the 1 $\mu$M and the 10 $\mu$M ChEMBL models, giving us two sets of predicted D-T associations. In order for a fair comparison to be made for each of the 2,223 drugs the top $x$[8] inferred single protein targets were collated and ranked. This process resulted in three sets of 215,075 ranked drug-target interactions; *DReS*, *Chem1* and *Chem10*.

### Target class comparison

We identify five human protein target classes based on their sizes and importance, as described by *Bull & Doig (2015)*. Proteins are classified as one of the following: G prote–incoupled receptors (GPCR); ion channels; kinases; proteases; and other. In order to do this we use the same approach described by *Bull & Doig (2015)*. Protein family membership is determined using multiple protein sources. The first is the id attribute of a `keyword` ($k$) element within a UniProt[9] entry $E$. All keywords assigned to $E$ are captured in the set $K$. If "KW-0297" in $E(K)$ then $E$ is classed as a GPCR; if "KW-1071", "KW-0851", "KW-0107", "KW-0869", "KW-0407", "KW-0631" or "KW-0894" is in $E(K)$ then $E$ is classed as an ion channel; if "KW-0418", "KW-0723" or "KW-0829" is in then $E(K)$ then $E$ is classed as a kinase; if "KW-0031", "KW-0064", "KW-0121", "KW-0224", "KW-0482", "KW-0645", "KW-0720", "KW-0788" or "KW-0888" is in $E(K)$ then $E$ is classed as a protease; and finally all other proteins are classed as 'other'. A protein is also classified as a GPCR, kinase or protease if it appears in the GPCR[10], kinase[11] or protease[12] files respectively.

## RESULTS

### Characterisation and performance of DReSMin

We evaluated the effectiveness of each step of our algorithm by adding each step (initial candidate set selection, topological pruning and semantic distance thresholding) sequentially to the basic topological search algorithm and then comparing the efficiency of each modified version to the VF2 topological search. The algorithm was implemented on a 20 node Ivy-Bridge bioinformatics cluster. Performance was measured as the time taken
for a complete search for a semantic subgraph ($Q$) within a given target graph ($G$). Random semantic target graphs ($Ran$) as well as random semantic subgraphs were produced in order to evaluate the performance of the semantic subgraph search strategy. These random graphs were formulated using an approach that attempted to replicate the semantic and topological properties of $Dat$. In these random target graphs $\forall v \in V(Ran)$ of type $t_v$, the average $deg^-(t_v)$ and the average $deg^+(t_v)$ were maintained $\forall t_v \in T_v(Dat)$. Experiments were repeated 10 times.

The SDC and graph-pruning step display linear running times of $O(n)$; with the former capable of scoring $8 \times 10^4$ concept pairs per second and the latter taking <1 second to prune a graph $G$, with $|V(G)|$ of $1 \times 10^6$. During the performance measures we focused on semantic subgraphs with between 3–6 vertices. The effect on search time when altering semantic subgraph edgeset size was also examined (Fig. S2) showing an improvement in performance as the edgeset size increases. This performance increase is due to the fact that fewer nodes satisfy the more stringent topological rules. With more stringent pruning during a run of the algorithm the search space at each state is reduced; ultimately meaning that when searching for semantic subgraphs who share the same $|V|$ but have differing $|E|$, the semantic subgraph with the $> |E|$ will be more efficient to search for.

Once semantic subgraphs reach a $|V(G)|$ of 4 then restricting the initial candidate set to include only `Compounds` improves performance. It is at this point the benefits of reducing the initial candidate set successfully reduce the search space, concomitantly increasing performance (Fig. 5). A similar phenomenon is observed with the introduction of the closed world check, whereby the real performance benefits are apparent when semantic subgraphs reach a $|V(G)|$ of 4 (Fig. 5). By restricting the initial candidate set as well as using the closed world assumption a two fold increase in performance in comparison to a purely topological approach was observed. Performance is further enhanced when utilising the semantic distance calculator demonstrating an almost 10 fold performance boost when comparing to the purely topological approach.

The semantic graph prune step introduces a small but noticeable increase in performance to DReSMin. Despite an overall increase in performance the graph prune step also brings a subtle cost; any potential matches containing an element that scores $<ST$ when passed to the SDC will not be returned. It is for this reason that the graph pruning step is an optional add-on to the DReSMin algorithm. The graph pruning step is most useful when one wishes to return matches that are semantically exact to the semantic subgraph being used as a query ($Q$). The graph split step can potentially reduce the search time for $Q$ from that of a $|V(Q)|$ of 6 to one closer to the sum of a search for a subgraph with a $|V(Q)|$ of 3 and a subgraph with a $|V(Q)|$ of 4. It is this step that produces the greatest improvement to performance. For example, when using the SDC to search for $Q$, where $|V(Q)| = 6$ in $G$ when $|V(G)| = 1 \times 10^5$, takes 60 seconds, using the graph split method reduces this search time to just over 8 seconds, a 7 fold increase in performance.

Overall, when using all three of the algorithmic steps in DReSMin, the performance of DReSMin showed performance characteristics approximating a linear scale closer to $O(n)$. This is in contrast to the exponential scaling characteristics observed for the purely topological search algorithm, VF2. These DReSMin performance

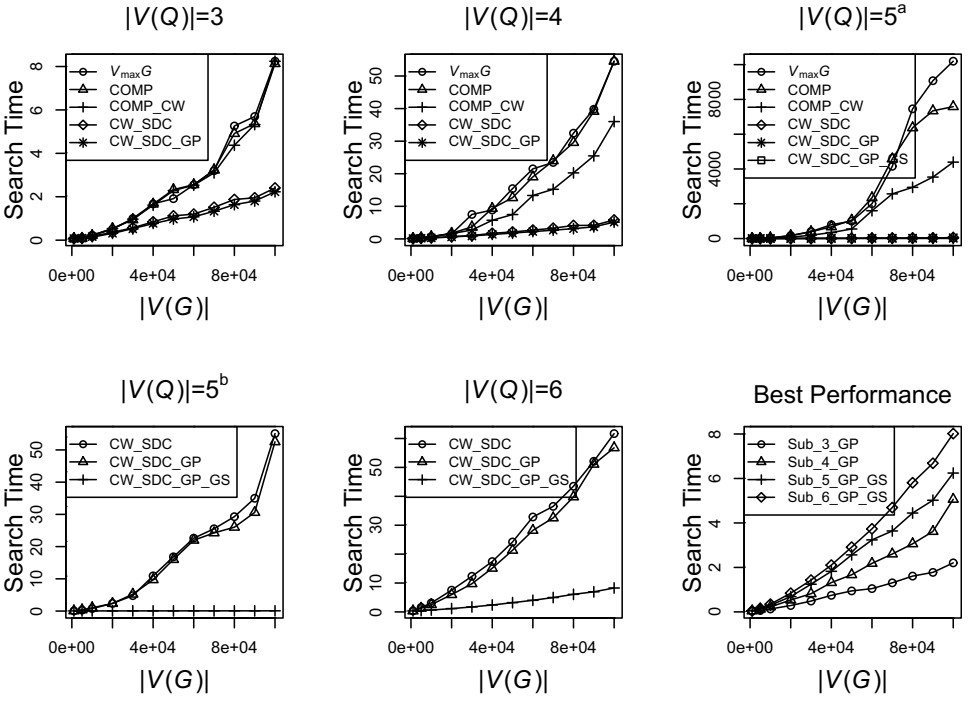

**Figure 5 Overview of algorithm performance with semantic subgraph (Q) queries nodeset |V(Q)| ranging from 3–6.** Graphs show each step of the algorithm reducing search time. Abbreviations: $V_{max}G$, initial candidate set is chosen using the $v_t$ of the node displaying the highest connectivity in $Q$; COMP, compound makes up the initial candidate set; CW, closed world check implemented; SDC, semantic distance calculator used during search; GP, semantic graph prune step implemented; GS, graph split used. $5^a$ (all steps plotted for $|V(Q)| = 5$) and $5^b$ (focuses on GP and GS of $|V(Q)| = 5$). Algorithm Performance (bottom right) shows the best approach for each semantic subgraph size.

characteristics were observed for semantic subgraphs of size $|V(Q)| \leq 6$ (Fig. 5). Using DReSMin with the hardware described above it is possible to complete an exhaustive, exact search for a 6 node semantic subgraph in a target graph containing $>1.5 \times 10^5$ vertices in under 10 seconds. The accuracy of the algorithm does not decrease as the target graph connectivity, or $|E|$, increases (Fig. S3) or as the target graph $|V|$ increases (Fig. S4).

## Application to search for drug-target interactions

Semantic subgraphs were identified in *Dat* and used to infer novel potential D-T interactions in *Dat* using the DReSMin algorithm. To aid in this process we utilised more recent D-T versions of the DrugBank datasets that were not used to build *Dat*. This approach allows us to determine if D-T interactions inferred from *Dat* using DReSMIn are likely to be supported as more knowledge is obtained. We can thus understand if inferences made have any potential value to drug repositioning now, as opposed to in the future.

To carry out this process the D-T interactions from DBv2.5 that were integrated into *Dat* were retrieved and captured in the set *DatRel*. We used DBv2.5 to construct *Dat* in this exercise even though later releases of DrugBank are available; v3.0 (DBv3) and v4.2 (DBv4.2) (*Knox et al., 2011*). DBv3 contains additional drugs, targets and their

**Table 1 Drug, Target and Drug-Target (D-T) Interactions present in the Dataset and DBv3.**

|  | Drug | Target | D-T Interaction | Unique | Relevant |
|---|---|---|---|---|---|
| *Dat* (DBv2.5) | 4,772 | 3,037 | 9,227 | – | – |
| DBv3 | 6,180 | 4,080 | 14,570 | 8,768 | 2,919 |
| DBv4.2 | 6,377 | 3,601 | 14,157 | 8,673 | 2,940[*] |

**Notes:**
Unique: refers to interactions not found in Dat, Relevant: subset of Unique interactions, whereby both the drug and target can be found in *Dat*.
[*]Of these 333 are unique to DBv4.2 (i.e they are not captured in DBv3).

interactions to those already contained in *Dat* (Table 1) and 8,768 additional D-T interactions to those found in *Dat*. Of these interactions, 2,919 involve drugs and targets that are present in *Dat*, but the interaction relationship had not yet been defined (i.e. the D-T interaction had not been annotated in DBv2.5). In this work, we refer to these 2,919 interactions from DBv3 as being 'relevant'. These relevant interactions are represented in the set *DBv3Rel* (see Eq. 5) and were used to derive a query set of semantic subgraphs that were in turn used to mine *Dat*. DBv4.2 was then used as a reference to validate the new repositioning opportunities identified through the mining of *Dat*.

$$DBv3Rel = \{DatRel \cup Unique(DBv3) \mid d \in DatRel(d) \wedge t \in DatRel(t)\} \tag{5}$$

### Semantic subgraphs inferring drug target interactions

Semantic subgraphs can be derived through manual exploration of the graph and by reference to known repositioning examples. However, in this work, in order to exhaustively test the DReSMin algorithm, we derived an automated method for producing a set of semantic subgraphs that would be appropriate for systematically mining for new D-T interactions. In order to produce such a set, we extracted the portions of the network in *Dat* that contained drugs and targets from the 2,919 D-T interactions whose interaction was annotated later in *DBv3Rel*. To extract the subnetworks, each drug and target pair was identified in *Dat* and the subnetwork represented by the shortest path between them was extracted as a semantic subgraph (Fig. 6). To identify the shortest semantic subpaths, *Dat* was converted to an undirected graph and a Java implementation of Dijkstra's shortest path algorithm (*Dijkstra, 1959*), from the JGraphT[13] library used. On carrying out this semantic subgraph identification exercise 194 different subgraphs with a $|V| < 10$ were found to cumulatively identify more than 95% of the D-T interactions in *DBv3Rel* and were used as a reference set for D-T inference using DReSMin as described below.

### Inference of novel drug-target interactions

The 194 semantic subgraphs were used as queries to search *Dat* using DReSMin to test the ability of the algorithm to identify D-T interactions in *Dat* that had not yet been annotated in DBv2.5 (but are present in DBv3). DReSMin was used to identify subgraphs in *Dat* that were similar to the query set of semantic subgraphs and therefore with the potential to be indicative of novel D-T interactions and ultimately aid in the identification

[13]http://jgrapht.org.

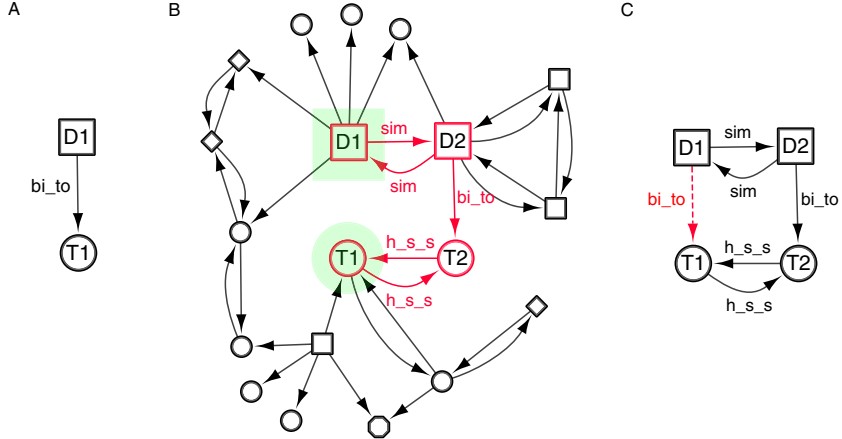

**Figure 6 Semantic subgraphs were derived from the semantic shortest paths between a drug and a target pair captured in DrugBank v3.** (A) shows a drug-target interaction captured in DBv3 made up of a drug (D1) and a target (T2) captured in our network, *Dat*. In order to create semantic subgraphs D1 and T2 are identified in *Dat* (highlighted in green in B) and the semantic shortest paths between the two nodes calculated (highlighted in red in B). Finally all semantic node types and edge types that fall on the semantic shortest path are used to create a query graph (C). *Note:* Dashed red line represents the inferred *binds_to* relations, squares represent compounds, circles targets, diamonds proteins and octagon diseases. For relation types: bi_to = *binds_to*, sim = *similar_to*, h_s_s = *has_similar_sequence*.

of drug repositioning opportunities. After an exhaustive search of *Dat* with the 194 semantic subgraphs a set of mappings (or instances) of each subgraph was identified. Semantic subgraphs were scored on their ability to identify D-T interactions captured in *DBv3Rel* (using Eq. 3), with these scores ranging from 0.0 to 0.06589 (Table S4). A single D-T interaction can be inferred by mappings of more than one query semantic subgraph, thus adding confidence to the prediction that a D-T interaction exists. Therefore, in order to rank the D-T interactions in terms of confidence, the scores assigned by all query semantic subgraphs that produced a mapping containing a potential D-T interaction were summed (using Eq. 4). The $R_q$ of the scores of all 194 query semantic subgraphs was 0.9499 (Fig. S5) and so inferred D-T interaction scores contained within mappings could potentially, range from 0.0 to 0.9459. The top ten performing subgraphs, and a larger illustrative subgraph, are shown in Fig. 7.

A search of *Dat* with the set of 194 semantic subgraphs described above resulted in 906,152,721 mappings. These mappings now captured the potential drug target interactions in the structure of the mapping subgraph. The 906,152,721 mappings predicted 9,643,061 D-T interactions that were ranked as described above. Unsurprisingly, we identify the interactions from *DBv3Rel* that were used to create the semantic subgraphs. Importantly, however, these interactions score highly, which indicates that a single interaction was identified by multiple semantic subgraphs. The D-T interactions from *DBv3Rel* consistently scored better and ranked higher than the unsupported inferred associations (Figs. 8A and 8B). We also observe that the D-T interactions subsequently annotated and captured in *DBv3Rel* are identified by two fold the number of semantic subgraphs that infer D-T associations not annotated and present in *DBv3Rel* (Fig. 8C).

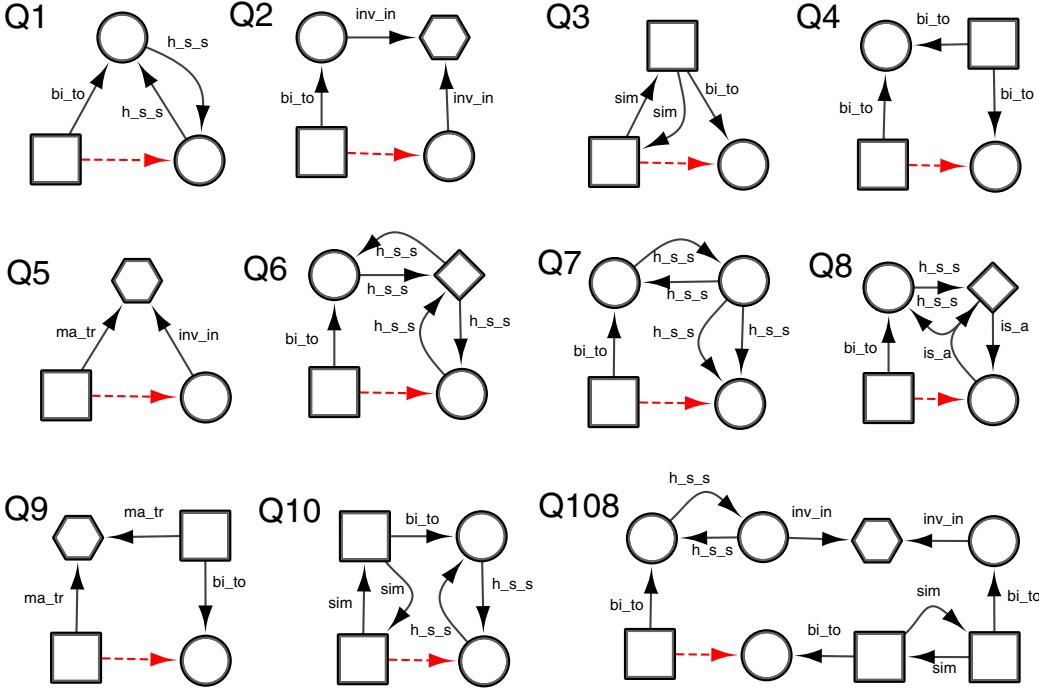

**Figure 7 Examples of semantic subgraphs drawn from the semantic shortest paths.** Q1–Q10 are drawn from the semantic shortest paths that represented the shortest path between the greatest number of D-T interactions in *DBv3Rel* and Q108 is an example of a more complex semantic subgraph. *Note:* Dashed red lines represent the inferred *binds_to* relations, squares represent compounds, circles targets, diamonds proteins and octagon diseases. For relation types: bi_to = *binds_to*, sim = *similar_to*, h_s_s = *has_similar_sequence*, ma_tr = *may_treat*, inv_in = *involved_in* and is_a = *is_a*.

However, in order to quantify the predictive power of DReSMin we examined how many of the high scoring D-T predictions were subsequently annotated in DBv4.2. DBv4.2 contains 333 interactions not captured in DBv3 or *Dat*. In this work, these interactions are represented in the set *DBv4Rel* (see Eq. 6). These 333 new interactions had not been used to construct the semantic subgraphs used for searching *Dat*. Of the 333 D-T interactions captured in *DBv4Rel*, 309 were successfully identified (94%). We also observed high ranking and scoring of 309 D-T interactions from *DBv4Rel* that were successfully identified by DReSMin (Figs. 8D and 8E). The average number of semantic subgraphs that have mappings inferring the 309 annotated D-T associations captured in *DBv4Rel* is increased >4 fold in comparison to the number of semantic subgraphs that produce mappings that infer interactions not captured in *DBv4Rel* (Fig. 8F).

$$DBv4Rel = \big\{ (DatRel \cup Unique(DBv4.2)) \cap DBv3Rel \mid$$
$$d \in DatRel(d) \wedge t \in DatRel(t) \big\} \tag{6}$$

Looking in more detail at the top 20 inferred D-T interactions (Table 2) we see 12 different drugs and eight targets. Drugs include: three antiarrythmic calcium channel blockers (Verapamil, Mibefradil and Bepridil); three phenothiazine antipsychotic agents

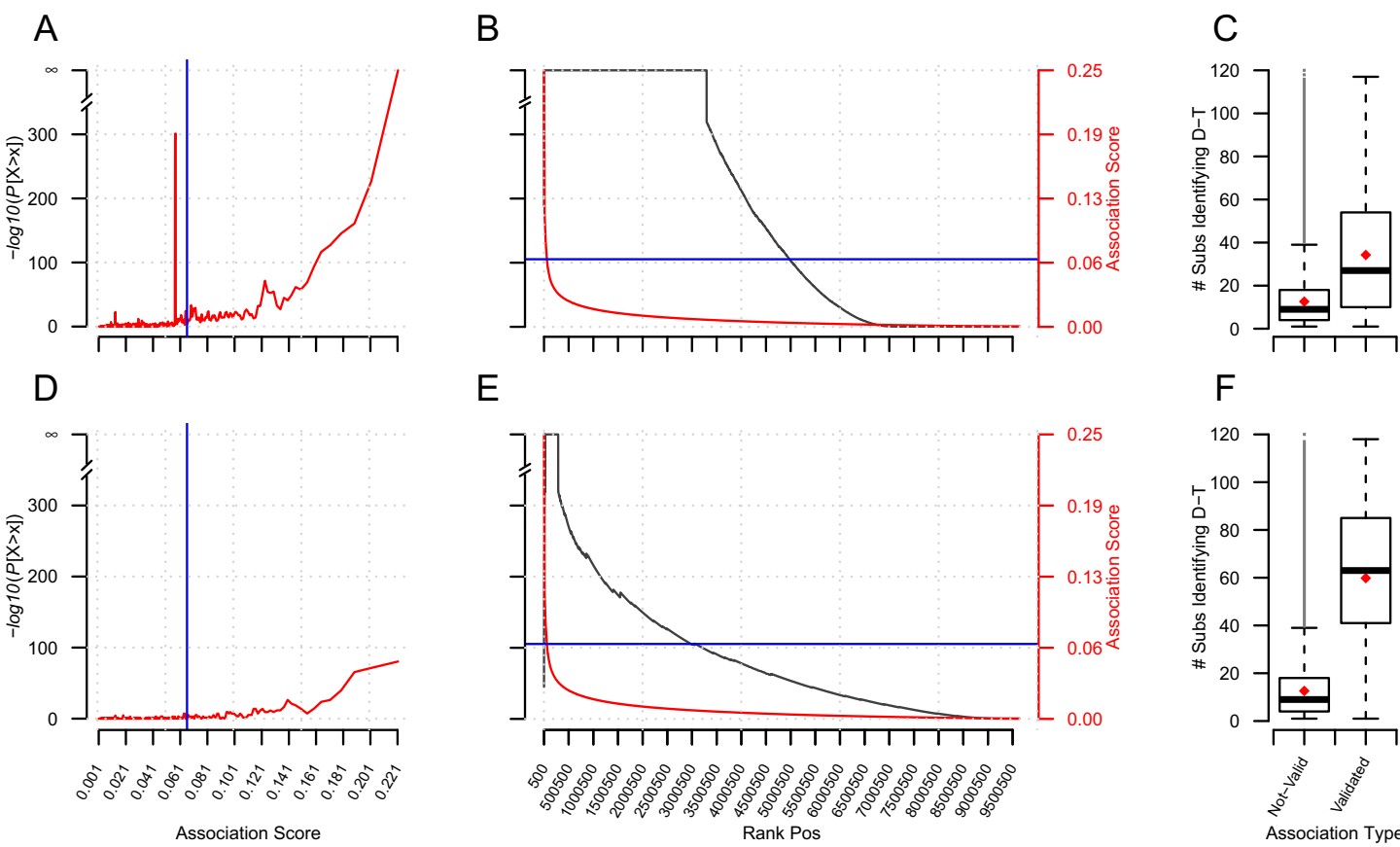

**Figure 8 Validation of inferred D-T associations with known D-T associations from DBv3 and DBv4.2.** (A, B and C) show how DReSMin identifies and ranks the 2,919 known interactions from DBv3 when searching *Dat*. (D, E and F) show how DReSMin identifies and ranks the 333 known interactions from DBv4.2. For (A and D) hypergeometric distribution of inferred knowns were calculated using the scores of the validated associations. For (B and E) hypergeometric distribution of inferred knowns were calculated using the ranked position of the validated interactions. (C and F) show the number of semantic subgraphs that inferred knowns in comparison to the number of semantic subgraphs that inferred novel interactions. *Note:* Blue line shows the highest scoring semantic subgraph; all scores above this line are definitely inferred by > one semantic subgraph.

(Promazine, Perphenazine and Thioridazine); three atypical antipyschotic agents (Propiomazine, Clozapine and Quetiapine); two anticonvulsants (Zonisamide and Levetiracetam) and one antiarrythmic adrenergic beta-antagonist (Propranolol). Of the 12 drugs captured in the top ranked inferred D-T interactions, the average number of D-T interactions captured in *Dat* is ~13, with the average number for all compounds being closer to three. The compounds present in the top 20 inferred D-T interactions are well studied and annotated and are thus highly connected in *Dat*. Targets include four voltage-dependant calcium channels (VDCC) and four G-Protein coupled receptors (GPCR). VDCCs display selective permeability to calcium ions which enter a cell, and alter a channel's properties, through the pore which is formed by the a 1 subunit. We can see that three sub-types of VDCC are represented in Table 2, being: L-type (CAC1C and CAC1D); P/Q Type (CAC1A) and N-type (CAC1B). Members of the GPCR superfamily in Table 2 include receptors activated by the neurotransmitters: serotonin (5HT7R and 5HT2B); epinephrine (ADA1A) and dopamine (DRD1).

**Table 2** DReSMin was executed using the 194 semantic subgraphs that represented the shortest path between the drugs and targets captured in the relevant associations.

| Drug (*DrugBank* ID) | Type, *Category* | Inferred Target (*Uniprot ID*) | Evidence | # Subs | Score |
|---|---|---|---|---|---|
| Verapamil (*DB00661*) | SM, *AP* | CAC1C (*Q13936*) | Y | 85 | 0.49211 |
| Mibefradil (*DB01388*) | SM, *WI* | CAC1A (*O00555*) | | 74 | 0.44378 |
| Mibefradil (*DB01388*) | SM, *WI* | CAC1B (*Q00975*) | | 59 | 0.43097 |
| Promazine (*DB00420*) | SM, *AP* | ADA1A (*P35348*) | Y | 117 | 0.39090 |
| Quetiapine (*DB01224*) | SM, *AP* | 5HT7R (*P34969*) | | 61 | 0.38779 |
| Propiomazine (*DB00777*) | SM, *AP* | 5HT7R (*P34969*) | | 69 | 0.38774 |
| Verapamil (*DB00661*) | SM, *AP* | CAC1A (*O00555*) | Y | 78 | 0.38436 |
| Verapamil (*DB00661*) | SM, *AP* | CAC1B (*Q00975*) | Y | 64 | 0.38180 |
| Mibefradil (*DB01388*) | SM, *WI* | CAC1D (*Q5SQC4*) | | 52 | 0.37525 |
| Perphenazine (*DB00850*) | SM, *AP* | 5HT7R (*P34969*) | | 86 | 0.37383 |
| Thioridazine (*DB00679*) | SM, *AP* | 5HT7R (*P34969*) | | 75 | 0.36830 |
| Promazine (*DB00420*) | SM, *AP* | 5HT7R (*P34969*) | | 75 | 0.36824 |
| Propranolol (*DB00571*) | SM, *AP, IN* | DRD1 (*P21728*) | | 96 | 0.36084 |
| Zonisamide (*DB00909*) | SM, *AP, IN* | CAC1B (*Q00975*) | | 50 | 0.35478 |
| Levetiracetam (*DB01202*) | SM, *AP, IN* | CAC1B (*Q00975*) | Y | 50 | 0.35478 |
| Thioridazine (*DB00679*) | SM, *AP* | 5HT2B (*P41595*) | | 107 | 0.35036 |
| Clozapine (*DB00363*) | SM, *AP* | 5HT7R (*P34969*) | Y | 64 | 0.34799 |
| Propranolol (*DB00571*) | SM, *AP, IN* | ADA1A (*P35348*) | | 84 | 0.34663 |
| Bepridil (*DB01244*) | SM, *AP, WI* | CAC1C (*Q13936*) | | 77 | 0.34610 |
| Levetiracetam (*DB01202*) | SM, *AP, IN* | CAC1A (*O00555*) | | 63 | 0.34605 |

**Notes:**
See Table S4 for a detailed scoring of the 194 semantic subgraphs. Scores are used as a ranking method for inferred interactions. Of the 20 interactions ranked highest by DReSMin, six were found in DBv3; having literature supporting their existence. For drug Type and *category*: SM, small molecule; *AP*, approved; *IN*, investigational and; *WI*, withdrawn. Scores are to 5 decimal places. # Subs refers to the number of semantic subgraphs that inferred the D-T interaction, with the maximum being 194.

The ability of DReSMin to predict novel D-T interactions was compared to the state-of-the-art ligand-based method from ChEMBL. We first examined how many D-T interactions were predicted by both methods (co-prediction) using interactions captured in the sets *DReS*, *Chem1* and *Chem10*. 35% of the top $x$ D-T interactions inferred by DReSMin are found in the top $x$ D-T interactions predicted by ChEMBL models (Figs. 9A and 9D). More interestingly is the fact that DReSMin successfully infers 10% more of the knowns from *DBv4Rel* than ChEMBL, for both models (Figs. 9C and 9F). We found that DReSMin is able to rank the known D-T interactions more effectively than ChEMBL, with a mean ranking position of known D-T interactions from *DBv4rel* of 16,977, as opposed to the 47,618 achieved by ChEMBL (47,746 for 1 uM model and 47,490 for 10 uM model). We must recognise the fact that the semantic subgraphs used during this work were derived using DrugBank data and the ChEMBL models trained on ChEMBL data.

### Target class comparison

After classifying all human proteins in *Dat* we identify: 826 GPCRs; 343 ion channels; 638 kinases; and 560 proteases. Of the 9,643,061 D-T interactions inferred by DReSMin, 4,780,935 (49.6%) involve human protein targets with 103 GPCRs; 85 ion channels;

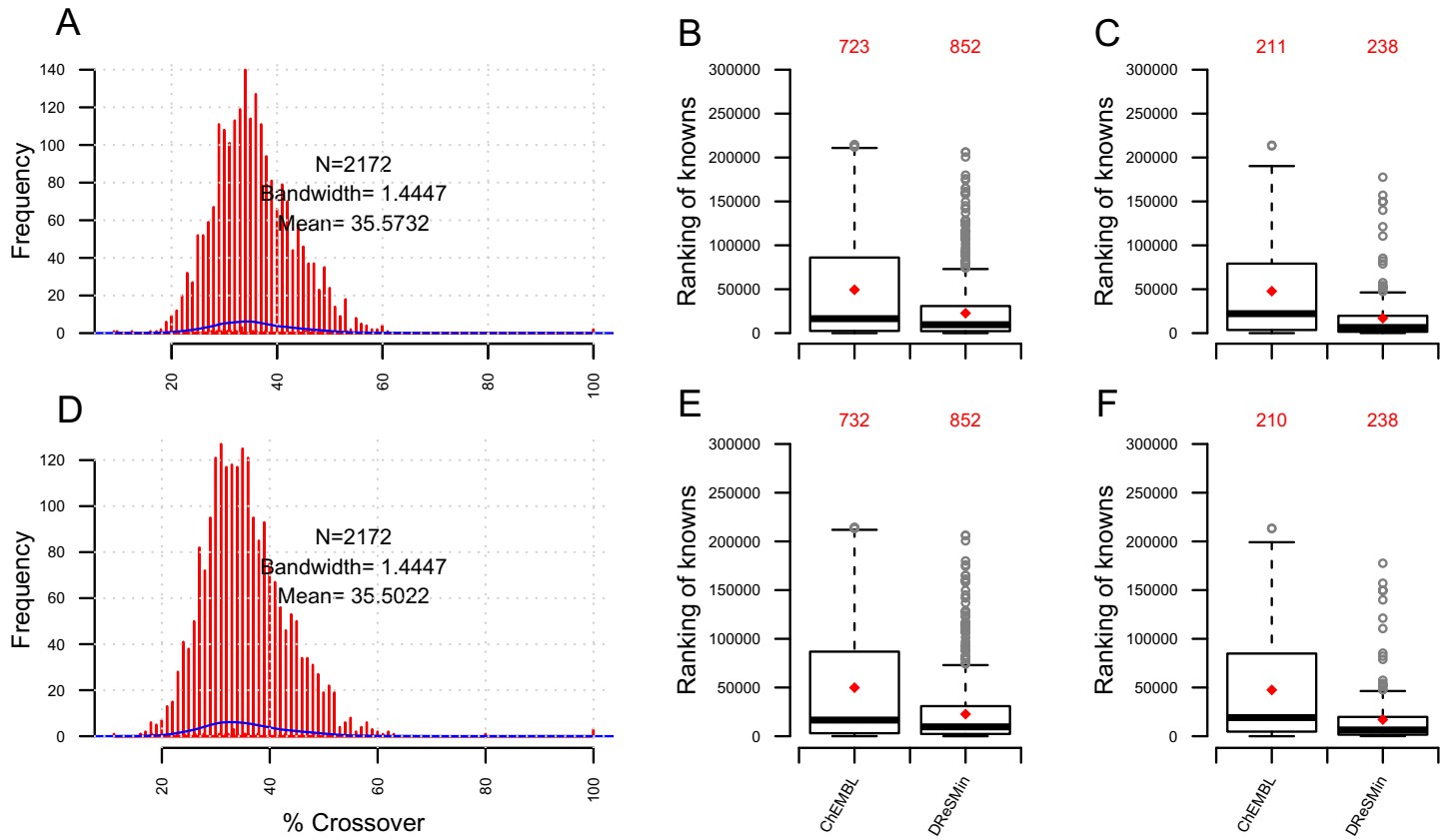

**Figure 9** **DReSMin inferred D-T associations in comparison to those inferred using the ligand-based similarity models provided by ChEMBL.** Top graphs (A, B and C) show comparison to those using the 1μM model from ChEMBL and bottom graphs (D, E and F) show comparison with the 10μM model. (A and D) show the % crossover between the top ranked $x$ associations from each method for each drug. (B and E) show the comparative ranking of the 2,919 known D-T interactions from *DBv3Rel*. (C and F) show the comparative ranking of the 333 known D-T interactions from *DBv4Rel*. In (B, C, E and F) red diamonds show the mean ranking and numbers in red show the number of knowns captured by each method. Only associations inferred by the 2,223 drugs with a mapping between DrugBank and ChEMBL and those that contain the overlapping set of target proteins are included in comparison.

89 kinases; 60 proteases; and 782 others are captured. In Fig. 10A we see how the target classes defined are represented in the inferences made by *DReSMin*. 89% of proteins are classed as 'other', and make up the targets in 69% of all inferences. GPCR's make up only 4% of all proteins and yet are involved in nearly 10% of all inferences. Only 3.2% of all proteins are classed as kinases, yet these are captured in 7.9% of all inferences. Proteases make up 2.8% of proteins and are shown to be part of 5.4% of all inferences. Finally, ion channels make up 1.7% of proteins and are contained in 7.4% of all predicted D-T associations.

Ion channels make up the second smallest set of comparative inferences, behind only proteases, however Fig. 10A shows us that this class is inferred by, on average: the highest scoring predictions (using Eq. 4); the highest ranked inferences; and the most semantic subgraphs per inference. Second is the GPCRs, followed by proteases, others and finally kinases.

Figure 10B shows how highly ranked the D-T interactions captured in *DBv3Rel* are ranked in each of the classes. D-T associations captured in *DBv3Rel* are ranked highest for

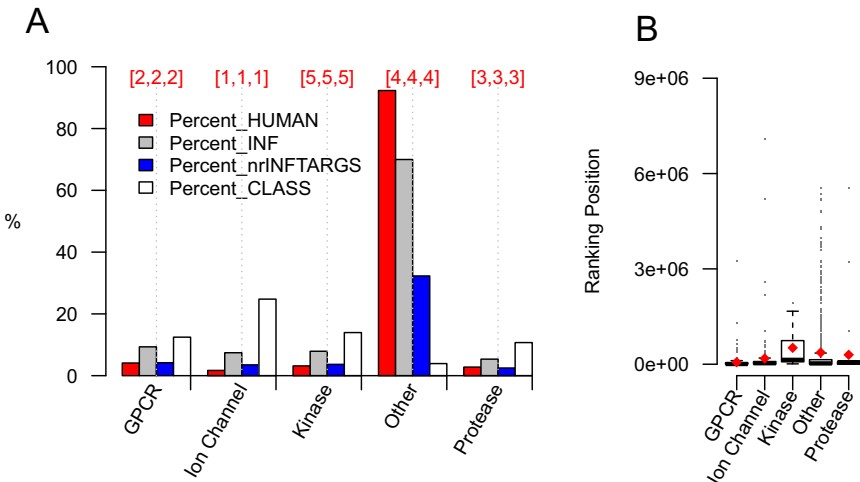

**Figure 10 Target classes.** Human proteins in *Dat* were assigned to one of five target classes: kinases, ion channels; G protein coupled receptors (GPCR); proteases; or other. (A) shows % of the total set of human proteins assigned to each class (Percent_HUMAN), what % of all DReSMin inferred associations contained a protein target from that class (Percent_INF), the % of the unique targets inferred by DReSMin that were from that class (Percent_nrINFTARGS) and the % of the target class for which an inference was made (Percent_CLASS). (B) shows how the known associations captured in *DBv3Rel* were ranked in the DReSMin inferred D-T interactions, with 1 being the highest ranked association. *Note:* Sets of numbers above each class in A represent how the target class ranked in performance in comparison to the other classes in the following measures: the average score of inferred interactions; the average ranking of inferences for that class; and the average number of semantic subgraphs that made an individual inference for that class. 1 represents the best performing target class and 5 the worst.

GPCRs, followed by: ion channels; proteases; other; and kinases. Known associations from all classes are, on average, ranked in the top 6% of all inferred associations.

### Completing the drug-target-disease pathway

The highest ranked D-T interaction identified by DReSMin, receiving a score of 0.49211, is supported by the literature and therefore known to the scientific community. This D-T interaction is between one of the antiarrythmic calcium blockers, Verapamil, and CAC1C. Within *Dat* eight indications are associated to Veparamil and 12 diseases associated to CAC1C. One indication, hypertension, shares a *has_Indication* association with Verapamil and a *involved_in* association with CAC1C. Although Verapamil is already used to treat hypertension, and the inferred D-T interaction already known, we see how DReSMin may be used to help understand the molecular mechanism of a drug and thus complete the 'drug-target-disease' pathway. Understanding the molecular mechanisms of drugs can only aid the identification of repositioning opportunities. In Fig. 11 we see examples of unsupported, and therefore novel, DReSMin inferred D-T interactions that also increase understanding of the molecular mechanisms involved in a drug's ability to treat a disease. Like Verapamil, Bepridil is also a calcium channel blocker with known antiarrhythmia activities. Used as a treatment of hypertension, we can see in Table 2 an inferred D-T association involving Bepridil and CAC1C. Bepridil is one of the two drugs from Table 2 that have been withdrawn from the market due to safety concerns. For this reason it is not

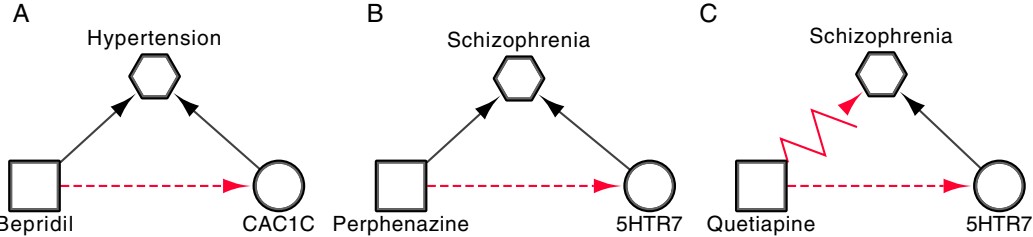

**Figure 11 Drug-target-disease pathways completed via inferred D-T associations.** Data presented is extracted from *Dat* with one association extracted from literature. *Note:* Dashed lines represent the inferred *binds_to* relations, zig-zag lines represent *has_Indication* relation not captured in *Dat* and extracted from literature, squares represent compound, circles target and octagon diseases.

a strong candidate to be repositioned, however, via the inferred association we are able to better understand the molecular mechanism of the drugs ability to treat hypertension (Fig. 11A).

In *Dat* we see three indications for Quetiapine (Psychotic Disorders, Bipolar Disorders and Autistic Disorders) and three *involved_in* associations involving 5HT7R (Schizophrenia, Pain and Muscular Diseases). Although a *has_Indication* association is not captured in *Dat*, Quetiapine is approved for the treatment of Schizophrenia. By integrating this knowledge with *Dat* and our inferred associations we can complete another drug-target-disease pathway (Fig. 11C). Schizophrenia, like many disorders, is a child of psychotic disorders. Although our dataset provides evidence showing Quetiapine is used to treat the parent term of Schizophrenia, psychotic disorders, the inference made allows for a better understanding of the drug-target-disease pathway in a more specific disease area to be achieved.

### Propranolol

One inferred D-T interaction in Table 2 involves the antiarrythmic adrenergic beta-antagonist, Propranolol, and the G protein-coupled receptor DRD1. In *Dat* we capture 12 indications for Propranolol and 17 disease associations for DRD1, with one indication, Hypertension, involved in both an indication association for Propranolol and a disease association for DRD1 (Fig. 12). Of the remaining 16 *involved_in* associations involving DRD1 three of the diseases represent known off-label indications for Propranolol being: Bipolar disorders; Schizophrenia, Alcoholism and as a non-stimulant treatment for ADHD (*Gobbo & Louzã, 2014*). The remaining 12 diseases present and support some interesting repositioning opportunities/studies of potential repositioning opportunities for Propranolol.

Looking at potential indications of Propranolol that are currently being investigated by the scientific community we see three that are supported by our work. *Dat* contains an association between DRD1 and cocaine related disorders, with multiple clinical trials being undertaken to analyse the use of Propranolol as a treatment for cocaine addiction (*NIDA, 2010*) as well as cocaine cravings (*Medical University of South Carolina, 2015*). A trial looking at the use of Propranolol as a treatment for Autism is also, at the time of writing, recruiting (*University of Missouri-Columbia, 2015*). Finally, a clinical trial has also

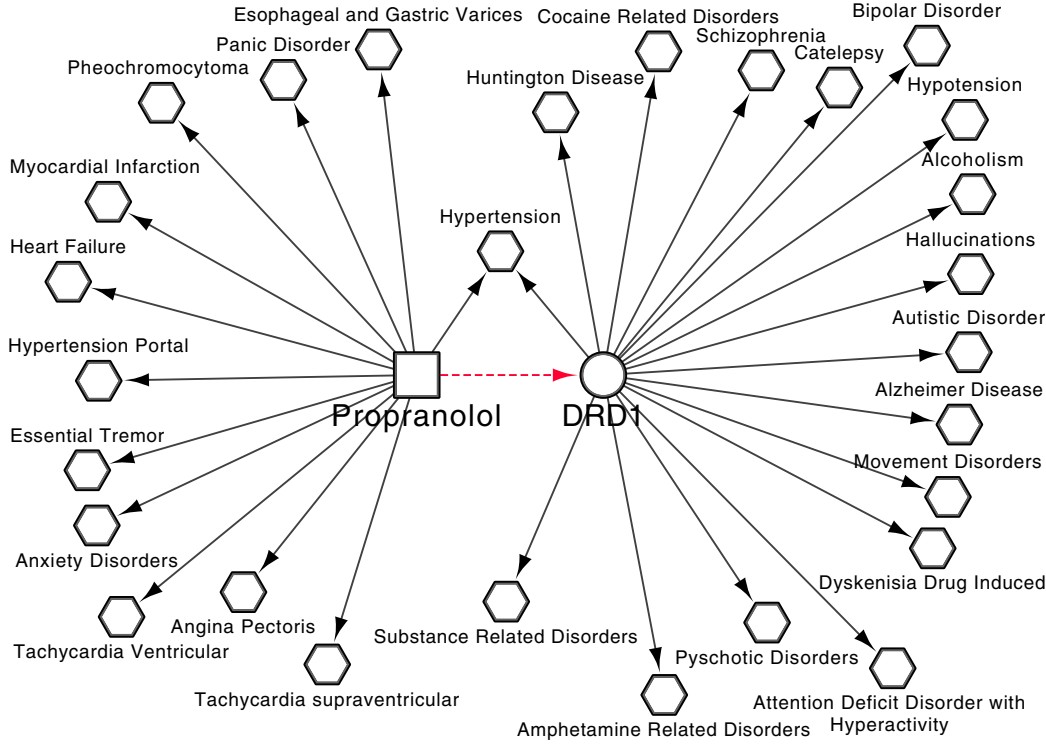

**Figure 12 Diseases associated with Propranolol and DRD1.** Drug-disease *has_Indication* associations involving Propranolol and gene-disease *involved_in* associations were extracted from *Dat. Note:* Dashed lines represent the inferred *binds_to* relations, squares represent compound, circles target and octagon diseases.

been undertaken to analyse the effects of using Propranolol as a treatment for drug-induced movement disorders (*Merck Sharp & Dohme Corp., 2014*). We can see that our inferred D-T interactions allow us to predict repositioning opportunities that agree with the community.

## DISCUSSION

In this paper, we explore the concept of using semantic subgraphs as a way of inferring novel D-T interactions with the aim of using them to identify drug repositioning leads. We present and formalise semantic subgraphs, showing how they may present patterns indicative of drug repositioning opportunities. By employing a novel approach to reducing the target graph size prior to a search, and by breaking larger semantic subgraphs to a set of smaller subgraphs, DReSMin significantly improves on the performance of a purely topological approach to pattern matching. We also show how the approach can be used to automate the identification of novel D-T interactions in an integrated semantic network, with the aid of historical data. This real-world problem often requires searching for semantic subgraphs where $|V(Q)| > 4$. The optimisations we have presented here makes searching for instances of these complicated subgraphs computationally tractable and scalable. We have shown an example of the application of DReSMin which highlights the potential of the approach.

When comparing DReSMin to other state-of-the-art drug-target prediction methods we observe an average co-prediction of 35%. Although this still leaves a large proportion of unique inferences we show that DReSMin inferences identify >16% of the knowns when using DBv3, and >10% of the knowns when using *DBv4*, in comparison to ChEMBL. The ability of DReSMin to identify more of the knowns than ChEMBL supports the approach and provides evidence that the approach provides an improved prediction set. We then directly compared and contrasted the results and found DReSMin to outperform the ChEMBL models at inferring annotated DrugBank D-T interactions. Considering DReSMin is a general algorithm, not specifically developed for the inference of D-T interactions, this highlights its potential ability to compete with specialised approaches. Although the semantic subgraphs used to search *Dat* were derived from the shortest paths between a drug and target from D-T interactions in DBv3, these interactions were inferred, on average, by around 40 different semantic subgraphs. This is in contrast to the 15 semantic subgraphs that inferred D-T interactions not captured in DBv3. Again this validates the approach we employed during this work. Annotated D-T interactions were not only captured by the semantic subgraph derived from the semantic shortest path between their drug and target but also by many more.

We show that DReSMin is able to identify known D-T interactions regardless of the class to which the target belongs. Having said this, the approach works better for target classes where there exists a relatively high amount of information, such as GPCRs and ion channels, whose knowns fall, on average, in the top 2% of inferred D-T interactions. Knowns involving other classes, such as kinases, for which there is less information, are still, on average, captured in the top 6% of all DReSMin inferred D-T associations. Our approach makes use of the holistic view of an entity and so if less is known about a target it will be captured in fewer semantic subgraphs and thus D-T interactions that it is predicted to be involved in will obtain a lower score. The data bias will become less of a problem as more and more data is produced for target classes such as proteases and kinases.

Although DReSMin at present scores semantics based purely on the most abstract form of types, it could be beneficial to include scoring metrics based on node and edge attributes, and the data-sources from which they are retrieved. For example, during the process of data integration it would be useful to consider dataset quality during the construction of the integrated graph and apply annotations that indicate a measure of confidence in a given interaction. To this end we are currently developing a new integrated dataset that will allow provenance and data to reliability to be scored during a search. This modification will allow the scoring of semantic subgraphs to be not only topological and semantic but also based on the reliability of the source of each element.

DReSMin is very capable of prioritising known D-T interactions, however, in order for inferences made to be useful to drug repositioning there are still some limitations that must be discussed. We illustrate these limitations with some examples. First of all, DReSMin infers an interaction between Dexrazoxane (*DB00380*) and Dactinomycin (*DB00970*) and the Sodium channel protein type 1 subunit alpha (*P35498*). This target is located predominantly in the brain and is heavily associated with epilepsy (*Mantegazza et al., 2010*; *Escayg & Goldin, 2010*). To reach the brain a drug must cross the blood-brain

barrier (BBB). Restricted by their pharmacokinetics, multiple drugs, such as Dexrazoxane (*DB00380*) and Dactinomycin (*DB00970*) (*Holm et al., 1998*), are unable to cross the BBB and so this inference is unlikely to highlight any realistic repositioning opportunity. Secondly, DReSMin infers D-T interactions involving drugs from a range of marketed states. Examples include drugs that have been withdrawn due to the fact that they are not as effective as first thought, such as Drotrecogin alfa (*DB00055*) in the treatment of sepsis, or due to poor sales, such as Halazepam (*DB00801*); interesting candidates for repositioning. Drugs that have been withdrawn from the market for reasons involving safety concerns prove a more problematic repositioning opportunity. Some examples are included in DReSMin inferences, such as drugs that have been withdrawn from market due to potentially fatal side effects, such as: Metamizole (*DB04817*); Grepafloxacin (*DB00365*); and Temafloxacin (*DB01405*). Finally, DReSMin inferred an association between Domperidone (*DB01184*) and the Beta-1 adrenergic receptor (*P08588*). Heart palpitations are a known side-effect of Domperidone and Beta-1 adrenergic antagonists, such as Propranolol have been administered to those suffering heart palpitations. One can thus deduce that Domperidone may have some agonistic action upon the Beta-1 adrenergic receptor. With these examples in mind other properties must be considered in further extensions to the approach. Drug properties, such as pharmacokinetics, in relation to the target location must be considered as well as a pre-filtering step to remove all drugs from the search that are likely unsafe. Post-filtering of results based on the likelihood of an D-T interaction prediction leading to a potential side-effect would also be a useful addition.

In the approach described here semantic subgraphs are derived from only the node types and edge types that fall directly on the semantic shortest path between a drug and a target. In order for a semantic subgraph capture even more functional detail it may be beneficial to expand the view that the subgraph takes of its immediate neighbourhood. To this regard we are currently considering extending semantic subgraphs to include nodes that interact with those in the semantic shortest path at a particular depth.

Although we present an exhaustive automated approach it is also worth noting that semantic subgraphs can be drawn from real life repositioning examples via manual curation. The manual development of semantic subgraphs, such as the one described in Fig. 1, is time consuming. However, manually curated semantic subgraphs may allow for more accurate representations of a functional module capturing a potential drug repositioning opportunity, as opposed to those created via automated approaches. We hope to create a library of semantic subgraphs curated from real world examples of repositioned drugs and compare the accuracy and efficiency to the semantic subgraphs developed during this work.

With regard to the mining algorithm, as new graph mining frameworks emerge with efficient graph searching algorithms (e.g. Neo4J), it may be possible to exploit these built in algorithms to implement sections of the approach we describe here. However, necessarily, the nature of these implementations will depend on the specific graph database.

We have demonstrated that our algorithm may be used to infer D-T interactions, however, like all *in silico* approaches to analysing in vivo and in vitro systems the accuracy is limited; overly simplified settings innately struggle to reflect real-life problems. Our approach, unlike many other computational approaches to drug repositioning, is not limited to the inference of D-T interactions. Semantic subgraphs may be designed to infer relations between any `conceptClasses` in a dataset and can be used to infer a drugs indication, mode of action, side effect and more. We believe that the systems biology approach that we describe here will allow for a more accurate, holistic, systematic approach to drug repositioning.

## ACKNOWLEDGEMENTS

We would like to thank Dr Philipe Sanseau of GlaxoSmithKline Research & Development Ltd for his guidance during this project. We also extend our gratitude to the ICOS Writing Group and acknowledge Bioinformatics Support Unit, both of Newcastle University for their discussions and manuscript input.

### Funding

JM receives funding as a CASE student from GSK and funding from the Engineering and Physical Sciences Research Council (ref 1592752). The funders had no role in study design, data collection and analysis, decision to publish, or preparation of the manuscript.

### Grant Disclosures

The following grant information was disclosed by the authors:
Engineering and Physical Sciences Research Council: 1592752.

### Competing Interests

Hannah Tipney and Peter M. Woollard are employees of GlaxoSmithKline.

### Author Contributions

- Joseph Mullen conceived and designed the experiments, performed the experiments, analyzed the data, wrote the paper, prepared figures and/or tables, reviewed drafts of the paper.
- Simon J. Cockell conceived and designed the experiments, analyzed the data, wrote the paper, reviewed drafts of the paper.
- Hannah Tipney conceived and designed the experiments, wrote the paper, reviewed drafts of the paper.
- Peter M. Woollard conceived and designed the experiments, wrote the paper, reviewed drafts of the paper.
- Anil Wipat conceived and designed the experiments, analyzed the data, wrote the paper, reviewed drafts of the paper.

## Data Deposition

https://bitbucket.org/ncl-intbio/dresmin

## Supplemental Information

Supplemental information for this article can be found online at http://dx.doi.org/10.7717/peerj.1558#supplemental-information.

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
