# Peer review of "Mining integrated semantic networks for drug repositioning opportunities"

_PeerJ, doi:10.7717/peerj.1558_

## Round 0.1 · original submission · Major Revisions

· Academic Editor

Major Revisions

Dear Joseph,

Thank you for submitting your manuscript for publication in PeerJ. It has been examined by three expert reviewers who have concluded that the work is well written and potentially sound and suitable for publication; however, it appears that at least two of the reviewers do not consider the limitations of your method adequately addressed. Especially reviewer 2 raises an important question that needs to be addressed in the revision thoroughly: If your approach produces 10% overlap of your results with others, and 90% is different, what makes you believe that the 90% difference supports your approach? In the present manuscript it seems that you were not yet able to convince all reviewers why your 90% difference is actually important and an improvement, which means that a revision will be needed prior to its further consideration for publication. In that respect I find reviewer 1's idea worth considering if not to present at least one example where the method does a prediction that is wrong, to aid understanding of the limitations.

·

Basic reporting

The authors present a novel method based on semantic subgraphs to infer drug-target interactions. This is a new and important field in chemo/bio-informatics and novel methods are needed to take advantage of the avalanche of data generated and integrated within the lifesciences.

Experimental design

The method is (to the reviewer's knowledge) novel, well descibed and the (in silico) experiments rigourosly done.

Validity of the findings

The results are interesting, sound and reasonable. However, the authors do not discuss the limitations of their method.It is expected that the methods would work very well for bioaminergic GPCRs and voltage-gated Ion-Channels due to the abundance of information for these target classes. It would be useful if the authors also investigates if the method could work succesfylly for other target classes like kinases and say peptidergic GPCRs where there is potentially less information. Also it would be useful with at least one example where the method does a prediction that is wrong, this would provide a better understanding of the limitations and improvement opportunities in the quality and integration of the underlying datasources as well as the applied algorithms.

Reviewer 2 ·

Basic reporting

No Comments

Experimental design

The approach seems to rely on directed links: This renders the entire system highly dependent on the data sources used, thus potentially limiting the general purpose usefulness, as a single set of biases (for example, when using any single source) is now replaced by the biases of the specific small set of sources used. The Authors can improve the paper by showing any inherent biases caused by the specific sources versus any random source in the field.

Validity of the findings

The Authors state that the ChEMBL approach is dissimilar (Line 413-415). They also state that in their comparison of DReSMin with other state-of-the-art target prediction methods, the average co-prediction is 10% (Line 473-474) . If state-of-the-art methods, now including the one offered here, only agree by 10%, is this not a problem? Simply stating that the methods are different, and thus it is to be expected that the results will be 90% different, severely limits the ability of a reader to place this work in its proper context. Here, the Authors seem to “gloss over” this issue. As a reviewer, I was disappointed to be asked to accept yet another 10% similar approach to other very similar ones. A proper discussion of the 90% difference, and whether this is important or not, should be added.

Additional comments

This is a paper that has been well-written, with excellent attention to detail by the Authors.

There are, however, some areas of potential improvement.

Line 16: “Drug repositioning is the application of established, approved compounds to a novel therapeutic application.” This is a limited and in some cases very mis-leading statement. DR also applies to drugs in development, not approved, shelved, or even at pre-clinical stages.

Line 30: The literature cited here is limited. The Authors should consider citing relevant foundational chapters in the first compiled book on the DR subject: http://www.wiley.com/WileyCDA/WileyTitle/productCd-0470878274.html “Drug Repositioning: Bringing New Life to Shelved Assets and Existing Drugs”.

Please also see commentary under Validity of Findings.

Reviewer 3 ·

Basic reporting

The paper seems well written. Below are my few concerns:

1. A semantic subgraph seems to be defined as a task in the paper:
“A semantic subgraph aims to infer a relation between vertices of a particular where a relation does not exist”
However, semantic subgraph should be a structure; the above statement refers to the link prediction task on semantic subgraphs.

2. Many related papers in drug-target interaction prediction domain are not covered in this paper. For example:
a.
An important work in this domain:
Yamanishi, Yoshihiro, et al. "Prediction of drug–target interaction networks from the integration of chemical and genomic spaces." Bioinformatics 24.13 (2008): i232-i240.
b.
A recent survey paper:
Ding, Hao, et al. "Similarity-based machine learning methods for predicting drug–target interactions: a brief review." Briefings in Bioinformatics 15.5 (2014): 734-747.
c.
A related work with similar structures to Figure1:
Fakhraei, Shobeir, et al. "Network-based drug-target interaction prediction with probabilistic soft logic." Computational Biology and Bioinformatics, IEEE/ACM Transactions on 11.5 (2014): 775-787.
d.
A related work in semantic similarity space:
Palma, Guillermo, Maria-Esther Vidal, and Louiqa Raschid. "Drug-target interaction prediction using semantic similarity and edge partitioning." The Semantic Web–ISWC 2014. Springer International Publishing, 2014. 131-146.

Experimental design

Seems valid, though have not checked thoroughly.

Validity of the findings

Seems valid, though have not checked thoroughly.

Additional comments

This paper explores the concept of using semantic subgraphs as a way of inferring novel Drug-Target interactions. The paper proposes an algorithm called DReSMin to detect them. The algorithm contains four components: Semantic graph pruning, Topological search, Semantic subgraph distance exclusion, Semantic subgraph splitting.
DReSMin employs an approach to reduce the target graph size prior to a search and breaks the larger subgraphs.
It is an interesting paper. However, the major technical emphasis and connection of the paper is to the graph mining literature. Making better connections and comparison to the related work in the drug-target interaction domain, would improve the strength of the paper.

---

## Round 0.2 · accepted · Accept

· Academic Editor

Accept

Dear Joseph,

Your revised manuscript has been re-examined by two of the three expert reviewers. Both reviewers feel that you have addressed their comments adequately. For the comments that were raised by the third reviewer I myself checked the manuscript carefully and consider the concerns of this reviewer addressed properly, too. Therefore I accept your article for publication.

·

Basic reporting

The authors has adressed my comments so I support publication

Experimental design

The authors has adressed my comments so I support publication

Validity of the findings

The authors has adressed my comments so I support publication

Reviewer 3 ·

Basic reporting

No Comments

Experimental design

No Comments

Validity of the findings

No Comments

Additional comments

The authors have properly addressed my concerns. The manuscript is more connected to the drug-target literature now.